# Effect of Equibiaxial Pre-Stress on Mechanical Properties Evaluated Using Depth-Sensing Indentation with a Point-Sharp Indenter

**DOI:** 10.3390/ma16020528

**Published:** 2023-01-05

**Authors:** Takashi Akatsu, Yoshihide Tabata, Yutaka Shinoda, Fumihiro Wakai

**Affiliations:** 1Faculty of Art and Regional Design, Saga University, 2441-1 Oono-otsu, Arita-cho, Nishimatsuura-gun, Saga 844-0013, Japan; 2Laboratory for Materials and Structures, Institute of Innovative Research, Tokyo Institute of Technology, R3-24 4259 Nagatsuta, Midori, Yokohama 226-8503, Japan; 3National Institute of Technology, Ube College, 2-14-1 Tokiwadai, Ube, Yamaguchi 755-8555, Japan

**Keywords:** depth-sensing indentation, pre-stress, elastic modulus, yield stress, piling-up, sinking-in, finite element method, elastoplastic, residual stresses

## Abstract

This study examined the effect of an imposed equibiaxial pre-stress (EBPS) on the evaluation of mechanical properties, using the depth-sensing indentation method with a point-sharp indenter, through a numerical analysis of indentations simulated with the 3D finite element method. The predicted elastic modulus, E^*^, and yield stress, Y^*^, were used as elastic and plastic deformation resistances under the indentation, respectively. It was found that both increased nominally with the increase in compressive EBPS and decreased with the increase in tensile EBPS, even though the induced change in the piling-up or sinking-in around the indentations was not significant. The effect of EBPS on E^*^ was described by the Hooke’s law for an isotropic elastoplastic material, whereas that on Y^*^ was accounted for by the change in the von Mises stress due to EBPS.

## 1. Introduction

Depth-sensing indentation is a technique in which local compressive-like properties (i.e., Young’s modulus and yield stress) and hardness can be evaluated by analyzing the relationship between the indentation load, P, and the penetration depth of an indenter, h (P–h curve, hereafter) [1,2]. Thus, indentation is often required to evaluate the mechanical properties of films and coatings on substrates. There are many studies on the evaluation of the mechanical properties of films/coatings on a substrate using the depth-sensing indentation method [3,4,5,6,7,8,9,10,11,12,13,14,15,16]. In these previous studies, the transitional change in the P–h curve, where the hardness and Young’s modulus varied from those of films/coatings to those of a substrate, was often discussed as a weight function of the maximum penetration depth, h_m_, which depends on the difference in the E, H, and Y between the film/coating and the substrate [3,9,14]. In particular, for films/coatings on a substrate system, the effect of the pre-stress owing to the mismatch in thermal expansion/shrinkage between the films/coatings and substrate [17,18,19] on the P–h curve requires attention. However, the effect has not been discussed frequently and clearly, because it is not easy to examine the transitional change and the pre-stress effect on the P–h curve simultaneously. In the case of the films/coatings on a substrate system, the equibiaxial pre-stress (EBPS) occurs in the vertical direction to the surface of the films/coatings. This paper focuses on the influence of the EBPS parallel to the indentation axis to the mechanical properties of bulk elastoplastic solids evaluated using the depth-sensing indentation method.

According to previous studies, where the effect of residual stress on the depth-sensing indentation was discussed [20,21,22,23,24,25,26,27], the nominal change in E and H due to the residual stress was entirely described by the change in the nominal contact depth, h_c_, which was underestimated due to enhanced piling-up around the indentation by the compressive pre-stress, while the h_c_ was overestimated through emphasized sinking-in by the tensile pre-stress. This indicates that E and H should be evaluated using the true contact depth under the pre-stresses. However, previous results are supported by elastoplastic solids with relatively small Y/E [20,21,22,24,25,26,27]. In such materials, plastic deformation was dominant under indentation, showing relatively large hysteresis in the P–h curve between loading and unloading [22,23]. Therefore, the effect of EBPS on the mechanical property evaluation using the depth-sensing indentation method should be assessed more systematically on wider range of Y/E values. Simulated indentation using the finite element method (FEM) has the advantage of analyzing the effect of EBPS on indentation by changing an elastic deformation-dominant material to a plastic deformation-dominant material [21,26,27,28,29,30,31,32].

In this study, the effect of EBPS on mechanical property evaluation using the depth-sensing indentation technique was systematically examined via indentations simulated with FEM. We have found an advantageous strategy to analyze elastic modulus, E^*^ [33], and yield stress, Y^*^ [34], which indicate elastic and plastic deformation resistances under the indentation, respectively, to clarify the effect of EBPS during indentation. Moreover, in the Appendix A, the approach for obtaining both deformation resistances E^*^ and Y^*^ without the influence of EBPS in the special case of thin films/coatings on a substrate is also discussed.

## 2. FEM Simulation of Indentation

A conical indentation on a cylindrical elastoplastic solid was modeled to simplify the modeling of a real pyramidal indenter. The 3D FEM simulation exploited the large strain elastoplastic capability of the ABAQUS code in the same way as reported in the literature [33,34,35]. Figure 1 shows the FEM model, where the 3D model is formed by the rotation of the 2D model, and the size of the mesh with relatively small aspect ratio becomes finer closer to the indentation in order to simulate P–h curve precisely. The validity required to simulate P–h curve with the FEM model has been confirmed in metals and ceramics through the comparison of the P–h curve between simulated and experimentally obtained [33,34,35]. The inclined face angle β of the rigid conical indenter was 19.7°, which is equivalent to that of the Vickers/Berkovich-type indenter. The friction between the indenter and surface of the elastoplastic solids was neglected for simplicity. The FEM simulation was performed using stress σ versus strain ε elastoplastic rules without strain hardening, which were σ=E ε for σ<Y and σ=Y for σ≥Y, for simplicity, although many elastoplastic solids show strain hardening. The effect of strain hardening on indentation is simply reflected as the increase in yield stress Y*, defined as $Y^{*}\equiv \frac{Y+E_{p}\varepsilon^{*}}{1-\left(\nu -b\right)}$, where Y is the yield stress, E_p_ is the plastic strain hardening modulus, ε* is the representative strain for point-sharp indentation, ν is the Poisson’s ratio, and b is a constant defined as b = 0.225tan^1.05^β with the inclined face angle of the indenter, β, [34]. The Young’s modulus, yield stress, and Poisson’s ratio of metals are often observed to be ~100 GPa, >1 GPa, and ~0.3, respectively. Then, indentations were simulated for E = 100 GPa, Y range = 1–15 GPa, and Poisson’s ratio (ν) = 0.3. von Mises criterion was used to determine the onset of the yielding flow. A constant displacement was forcibly applied to the circumferential side surface of the cylindrical solid to increase EBPS from −2 to 2 GPa before the indentation. Thus, EBPS was increased vertically in the direction of the indentation.

## 3. Results

### 3.1. Effect of EBPS on a P–h Curve

Figure 2 shows the simulated P–h curve of an elastoplastic solid with E = 100 GPa, Y = 3 GPa, and ν = 0.3. The solid line in Figure 2 shows the P–h curve without the influence of EBPS (σ_p_ in Figure 2). Compressive EBPS shifted the P–h curve toward increasing hardness, as shown by the circle markers in Figure 2, whereas the tensile EBPS shifted the curve toward decreasing hardness, as shown by the triangle markers. The predicted shift in the P–h curve due to EBPS was reported in the simulation of a previous study [20,36].

In the case of point-sharp indentation, P is usually given as a linear function of h^2^ according to the geometry self-similarity. The linear P–h^2^ relationship was not affected by the EBPS. Then, the indentation loading parameter k_1_ is obtained from the P–h curve for loading as $k_{1}\equiv \frac{P}{h^{2}}$. Figure 3 shows the nominal loading parameter, k_1n_, obtained in the presence of EBPS normalized by the EBPS-free loading parameter, k_1_. According to Figure 3, k_1n_/k_1_ increased with the increase in compressive EBPS (absolute value of minus σ_p_), whereas it decreased with the increase in tensile EBPS. The degree of the increase and decrease in k_1n_/k_1_ is more significant for plastic deformation-dominant solid with small Y/E values, shown with circle markers in Figure 3.

Figure 4 shows the ratio of the nominal to the EBPS-free dimensionless residual depths, ξ_n_/ξ. The relative residual depth, ξ, was defined as $\xi \equiv \frac{h_{r}}{h_{m}}$, with the residual depth, h_r_, being unaffected by EBPS. Note that the nominal ξ_n_ can be obtained using h_r_ affected by EBPS. According to Figure 4, ξ_n_/ξ decreased with the increase in compressive EBPS, whereas it increased with the increase in tensile EBPS. The degree of change in ξ_n_/ξ as a function of EBPS was determined to be independent from the Y/E value of the indented solid (Figure 4). 

### 3.2. Effect of EBPS on Piling-Up and Sinking-In around an Indentation

Figure 5 shows γ, which is defined as $\gamma \equiv \frac{h}{h_{c}}$, as a function of ξ. Piling-up around an indentation is represented with a large γ value, whereas sinking-in is represented with a small γ value. According to Figure 5, γ decreased with the increase in ξ, and the indentation transitioned from elastic to plastic. According to a previous study [34], γ, shown as a dashed line in Figure 5, is expressed as
(1)$$\gamma =\gamma_{e}\left(1-0.310{\;\gamma }_{e}\xi^{\frac{1}{0.310{\;\gamma }_{e}}}\right)$$
(2)$${\;\gamma }_{e}=1.56+0.208{\left(\nu -0.5\right)}^{2}$$
where γ_e_ is the γ value for a perfectly elastic solid. In Figure 5, γ slightly decreased with the increase in compressive EBPS (absolute value of minus σ_p_, filled markers), whereas it increased only slightly with the increase in tensile EBPS (unfilled markers). This indicates that compressive EBPS enhances piling-up, whereas tensile EBPS emphasizes sinking-in. However, γ changed only slightly owing to EBPS. This suggests that the significant changes in the P–h curves induced by EBPS (Figure 2, Figure 3, and Figure 4) are not attributed to the slight change in γ.

## 4. Discussion

### 4.1. Nominal Change in the Elastic Deformation Resistance E^*^ Due to EBPS

According to a previous study [33], where the elastic deformation resistance under the indentation was examined for elastoplastic solids with FEM, E^*^ is defined as $E^{*}\equiv \frac{E}{1-{\left(\nu -b\right)}^{2}}$, where $b=0.225\tan^{1.05}\beta$. $E^{*}$ can be evaluated using the P–h curve as
(3)$$E^{*}={\mathrm{ak}}_{e}$$
where k_e_ is an indentation elastic parameter defined as $k_{e}\equiv \frac{P}{h^{2}}$ for a perfectly elastic solid, and $a=1.31\tan^{0.919}\beta$. k_e_ for an elastoplastic solid is estimated as
(4)$$k_{e}=\frac{1-\xi }{1+1.84\xi^{1.32}}k_{2}$$
where k_2_ is the indentation unloading parameter defined as $k_{2}\equiv \frac{P}{{\left(h-h_{r}\right)}^{2}}$. Therefore, k_2_ can be expressed as follows:(5)$$k_{2}=\frac{k_{1}}{{\left(1-\xi \right)}^{2}}$$

Figure 6 shows the nominal E^*^ value, E^*^_n_, normalized by E^*^ without the influence of EBPS as a function of normalized EBPS, σ_p_/σ^*^, where σ^*^ is the representative indentation compressive stress. E^*^_n_ was derived using Equations (3)–(5) with nominal k_1n_ and ξ_n_ shown in Figure 3 and Figure 4. Here, σ^*^ is assumed to be the mean pressure under the indentation:(6)$$\sigma^{*}=\frac{\gamma^{2}}{\pi }k_{1}\tan^{2}\beta$$

According to Figure 6, E^*^_n_/E^*^ slightly increased with the increase in normalized compressive EBPS, as shown by the increase in the absolute value of minus σ_p_/σ^*^, whereas it decreased with the increase in normalized tensile EBPS. The dashed line in Figure 6 was drawn with Equation (11) as follows.

According to the Hooke’s law for an isotropical uniform elastic solid, the three-dimensional relationship between σ and ε in the diagonal components is given by
(7)$$\begin{array}{c} \varepsilon_{11}=\frac{1}{E}\left\{\sigma_{11}-\nu \left(\sigma_{22}+\sigma_{33}\right)\right\}\\ \varepsilon_{22}=\frac{1}{E}\left\{\sigma_{22}-\nu \left(\sigma_{33}+\sigma_{11}\right)\right\}\\ \varepsilon_{33}=\frac{1}{E}\left\{\sigma_{33}-\nu \left(\sigma_{11}+\sigma_{22}\right)\right\} \end{array}$$
where the indices of σ and ε represent the axes of coordinates, “1” corresponds to the direction of the indentation, and “2” and “3” correspond to the other coordinate directions. The elastic stress and strain under the indentation affected by EBPS is simply assumed as
(8)$$\varepsilon_{11}=\varepsilon^{*}$$
(9)$$\sigma_{11}=\sigma^{*}$$
(10)$$\sigma_{22}=\sigma_{33}=\sigma_{p}$$
where ε^*^ is the representative indentation compressive strain. Thus, the following equation can be obtained:(11)$$\frac{E^{*}{}_{n}}{E^{*}}=1-2\nu \frac{\sigma_{p}}{\sigma^{*}}$$
where $E=E^{*}{}_{n}$ and $\frac{\sigma^{*}}{\varepsilon^{*}}=E^{*}$ in Equation (7).

As shown in Figure 6, the relatively good agreement between E^*^_n_/E^*^ as a function of σ_p_/σ^*^ and the dashed line drawn using Equation (11) at ν = 0.3 indicates that the change in E^*^_n_ due to EBPS can be described by the effect of EBPS on the three-dimensional Hooke’s law.

### 4.2. Nominal Change in the Plastic Deformation Resistance Y^*^ Due to EBPS

According to a previous study [34], where the plastic deformation resistance under the indentation was examined for elastoplastic solids with FEM, Y^*^ is defined as $Y^{*}\equiv \frac{Y}{1-\left(\nu -b\right)}$ for an elastoplastic solid without strain hardening, which can be evaluated using the P–h curve as follows:(12)$$Y^{*}=\frac{1.37}{{\left\{\frac{1}{{\left(1-\xi \right)}^{\frac{3}{2}}\left(1-0.930\xi^{0.350}\right)}-1\right\}}^{\frac{2}{3}}}E^{*}\tan^{1.2}\beta$$

Figure 7 shows the nominal Y^*^ value, Y^*^_n_, normalized by Y^*^ without the influence of EBPS as a function of the normalized EBPS, σ_p_/Y^*^. Y^*^_n_ was derived using Equation (12) with nominal ξ_n_ and E^*^_n_ shown in Figure 4 and Figure 6. A dashed line in Figure 7 was drawn with Equation (16) as follows.

The von Mises stress, σ_M_, for an isotropically uniform elastoplastic solid is given as
(13)$$\sigma_{M}=\sqrt{\frac{{\left(\sigma_{11}-\sigma_{22}\right)}^{2}+{\left(\sigma_{22}-\sigma_{33}\right)}^{2}+{\left(\sigma_{33}-\sigma_{11}\right)}^{2}}{2}}$$

If EBPS is substituted as shown in Equations (9) and (10), the equation for σ_M_ becomes
(14)$$\sigma_{M}=\sigma^{*}-\sigma_{p}$$

This suggests that σ_M_ for indentation with EBPS is a representative indentation compressive stress shielded by EBPS. According to the von Mises yield criterion under indentation, σ_M_ is correlated with Y^*^_n_ using the constrained factor C [37,38,39] as
(15)$$\sigma_{M}={\mathrm{CY}}^{*}{}_{n}$$

Combining Equations (14) and (15) yields ${\mathrm{CY}}^{*}{}_{n}=C\left(Y^{*}-\sigma_{p}\right)$; thus, the equation becomes
(16)$$\frac{Y^{*}{}_{n}}{Y^{*}}=1-\frac{\sigma_{p}}{Y^{*}}$$

As shown in Figure 7, the relatively good agreement between Y^*^_n_/Y^*^ as a function of σ_p_/Y^*^ and the dashed line drawn using Equation (16) indicates that the change in Y^*^_n_ with EBPS can be described by the effect of EBPS on the von Mises stress for the elastoplastic solid.

The changes in E^*^ and Y^*^ can be described by the effect of EBPS on the Hooke’s law and the von Mises stress, respectively. This indicates that only the nominal values of E^*^_n_ and Y^*^_n_ affected by EBPS can be evaluated using the depth-sensing indentation technique if EBPS acts on the solid indented. Contrary to a previous study [29], it is impossible to distinguish E^*^_n_ and Y^*^_n_ from E^*^ and Y^*^, which should be obtained through depth-sensing indentation on a pre-stress-free solid, using the P–h curve obtained by a single indentation due to the lack of information. In particular, for indentation on very thin films/coatings on a substrate, E^*^ and Y^*^ can be estimated by comparing the indentation on the surface and on the cross-section (see Appendix A).

## 5. Conclusions

The effect of EBPS on the evaluation of E^*^ and Y^*^ using the depth-sensing indentation technique was examined by simulated indentations using FEM. E^*^_n_ and Y^*^_n_ increased with an increase in compressive EBPS, whereas they decreased with an increase in tensile EBPS, even though γ was not significantly affected by EBPS. The dependence of E^*^_n_ on EBPS was described by the three-dimensional Hooke’s law for an isotropic elastoplastic material, whereas the dependence of Y^*^_n_ on EBPS was depicted by the change in the von Mises stress due to EBPS.

## Figures and Tables

**Figure 1 materials-16-00528-f001:**
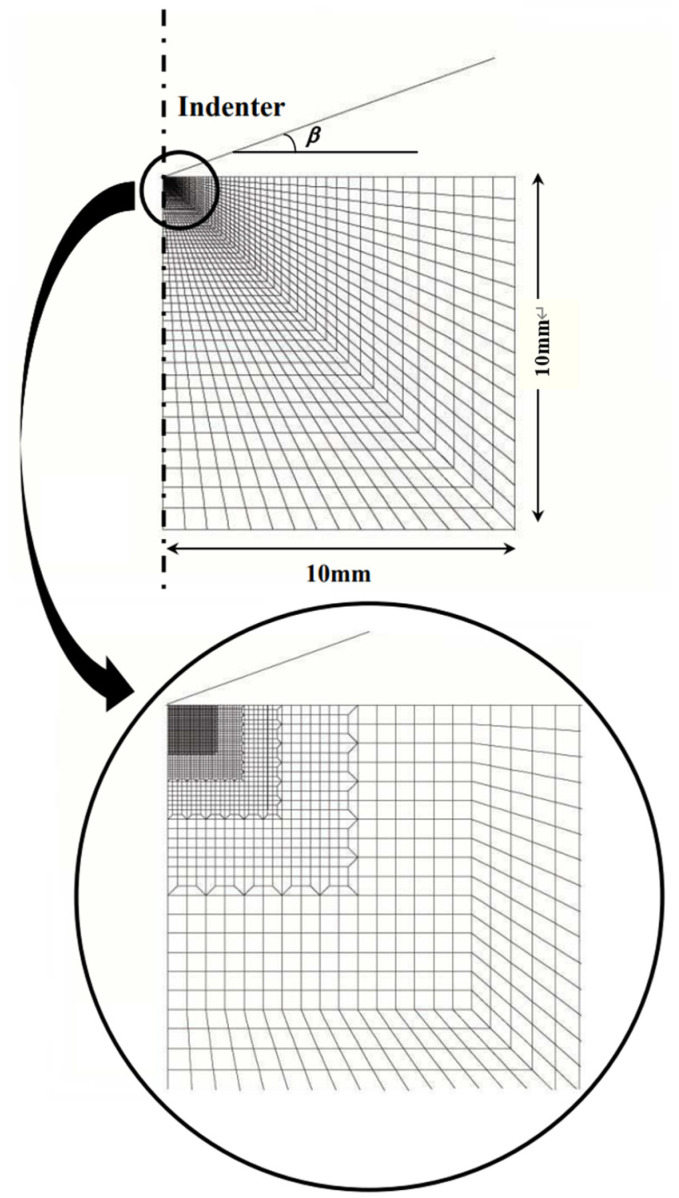
Detail of the FEM model geometry adopted in this study.

**Figure 2 materials-16-00528-f002:**
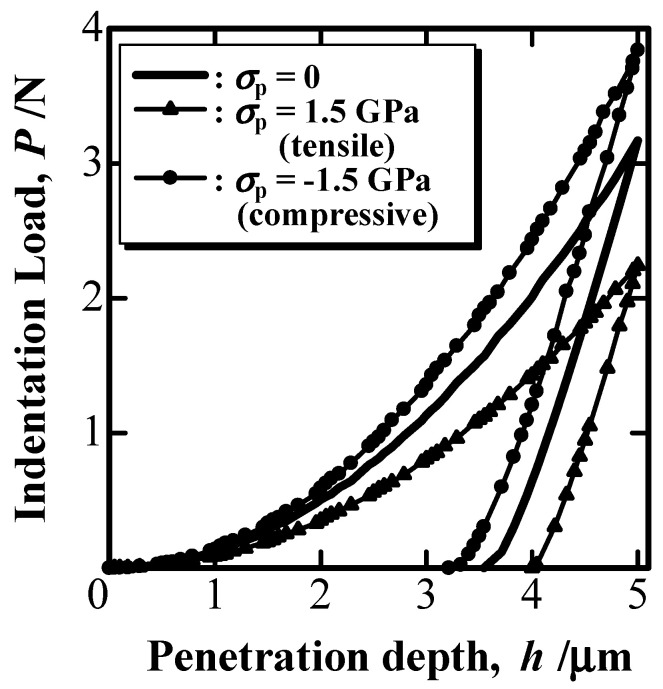
Simulated P–h curve affected by EBPS.

**Figure 3 materials-16-00528-f003:**
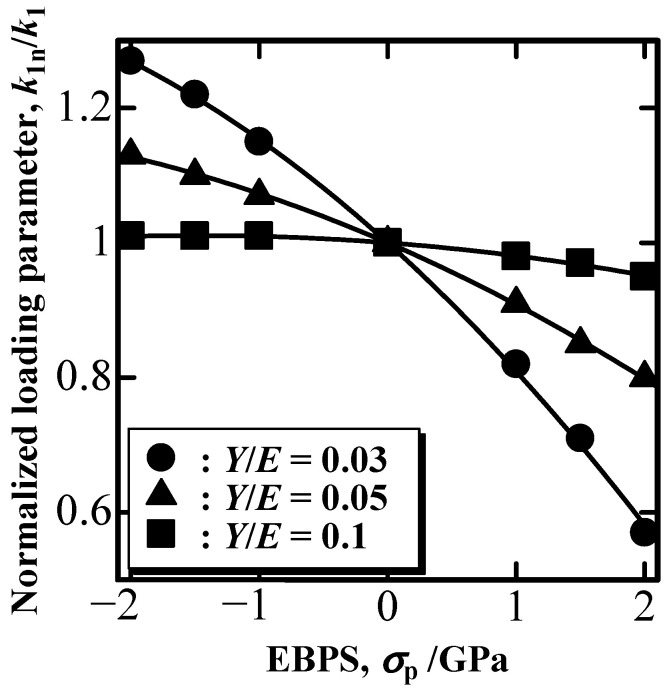
Normalized loading parameter, k_1n_/k_1_, as a function of EBPS.

**Figure 4 materials-16-00528-f004:**
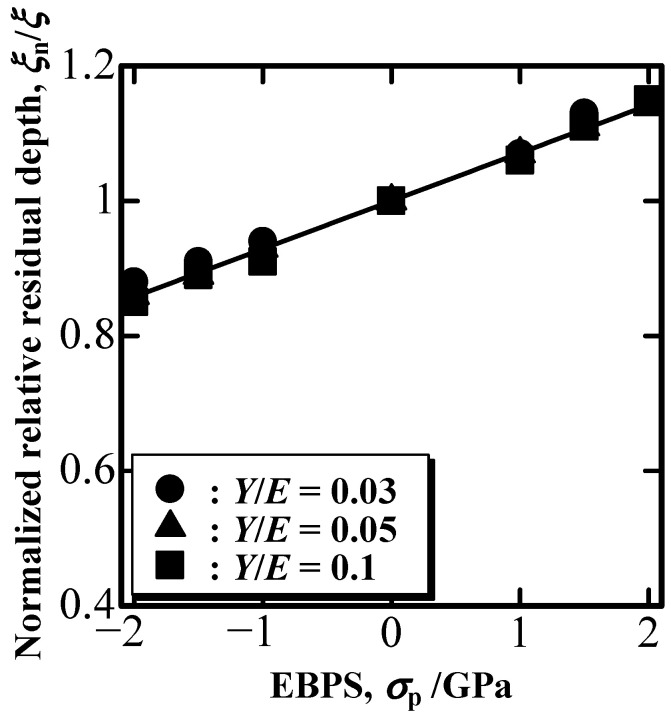
Nominal to EBPS-free relative residual depth ratio, ξ_n_/ξ, as a function of EBPS.

**Figure 5 materials-16-00528-f005:**
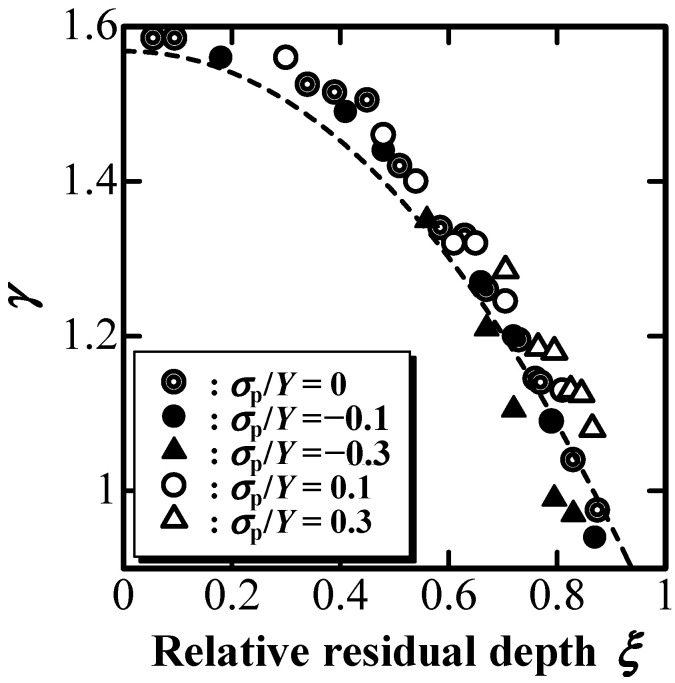
γ affected by EBPS as a function of ξ.

**Figure 6 materials-16-00528-f006:**
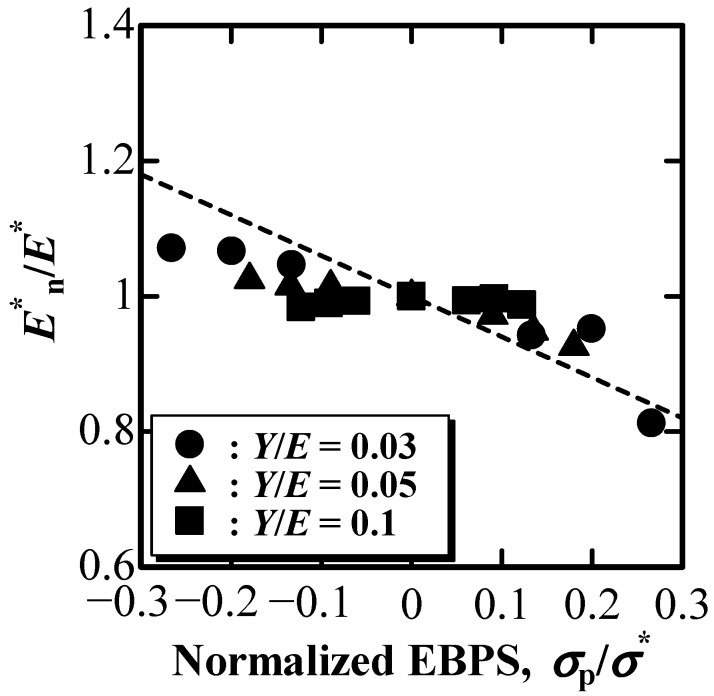
E^*^_n_/E^*^ as a function of normalized EBPS, σ_p_/σ^*^.

**Figure 7 materials-16-00528-f007:**
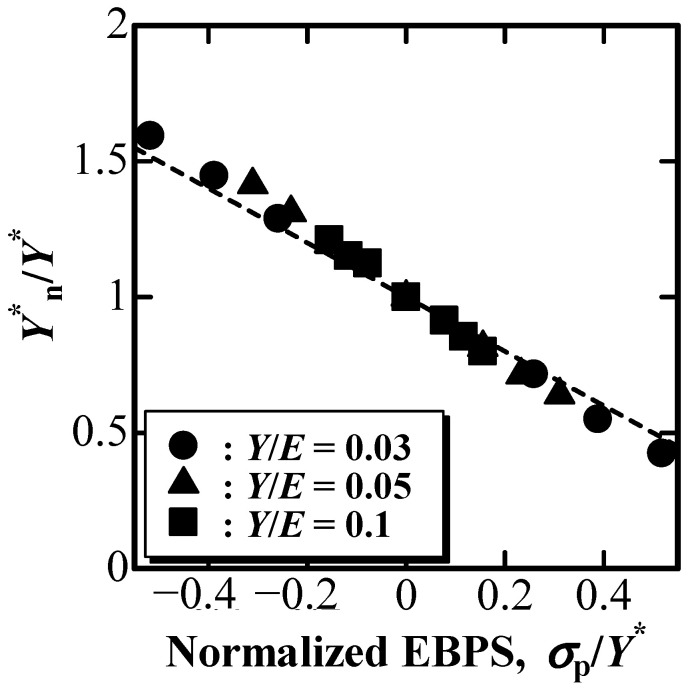
Y^*^_n_/Y^*^ as a function of normalized EBPS, σ_p_/Y^*^.

## Data Availability

Data is basically contained within the article.

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
