# Peer review of "Effect of Equibiaxial Pre-Stress on Mechanical Properties Evaluated Using Depth-Sensing Indentation with a Point-Sharp Indenter"

_materials, 2023, doi:10.3390/ma16020528_

Round 1
Reviewer 1 Report
Some line by line comments:
Line 87 “...EBPS (pre in Figure 1). Compressive EBPS shifted the P-h curve toward....”
How that stress is loaded??????? What it means, fisically???? Residual stress???
Line 112 “...Normalized elative residual depth, n/, as a function of EBPS…”
Must be “...relative…”
Line 114 “...Figure 4 shows γ@ which is defined as γ≡hhc, as a function…”
What is γ@?
Author Response
The authors really appreciate for advisory and meaningful comment given by the academic editor and the reviewers. The authors basically made revisions on the manuscript by using red characters according to the comment. The reviewer can get the reply to the reviewer's comment if the reviewer check the attached file.

Reviewer 2 Report
This manuscript shall be rejected outright, given that the work repeats old results with no new methods, theoretical ideas or new experiments. This is a classic undergraduate-level calculation, that was probably a little undergraduate project in the context of this manuscript, that repeats the results of Reference 20, which contains original calculations, using identical methods and focus. The results of Ref.20 have been repeated numerous times by researchers thereafter. The authors shall complement their calculations with molecular dynamics or dislocation dynamics calculations, or experiments, and add them in the manuscript, so that the work may render itself in a publishable form.
Author Response
The authors really appreciate for advisory and meaningful comment given by the academic editor and the reviewers. The reviewer can get the reply to the reviewer's comment if the reviewer check the attached file.

Reviewer 3 Report
This manuscript describes a study on depth-sensing indentation technique. This can be interesting information for readers but some questions should be checked.
Regarding the following (if I was able to understand correctly):
1. The review of background is not comprehensive and detailed enough. For example:Line37, the research focus and characteristics of references 3-16 should be reviewed separately or every 2-3 pieces. And the previous studies should be supplemented.
2. Line 73: In addition to directly citing references(33-36), the method description of FEM model should be described rather than directly referenced.
3. Line 76: As we all know that most of the materials are elastoplastic with strain hardening. What is the necessity of selecting the σ-ε rule as elastoplastic rule without strain hardening?
4. Fig.1: What is the date from of the circle and triangle markers? Is that simulation results? Please noted.
5. Line151,the assumption of Eq (9)-(10):How could σ* be assumed as σ11? The σ* was defined as “mean pressure” in Eq (7) and that was local stress under indentation.
Author Response

(The authors gave the same response as above.)

Round 2
Reviewer 3 Report
Thanks for your revision.
Author Response
The authors really appreciate for the reviewer's comment on our paper.